# Efficacy of Antimicrobial Photodynamic Therapy Mediated by Photosensitizers Conjugated with Inorganic Nanoparticles: Systematic Review and Meta-Analysis

**DOI:** 10.3390/pharmaceutics14102050

**Published:** 2022-09-26

**Authors:** Túlio Morandin Ferrisse, Luana Mendonça Dias, Analú Barros de Oliveira, Cláudia Carolina Jordão, Ewerton Garcia de Oliveira Mima, Ana Claudia Pavarina

**Affiliations:** 1Department of Dental Materials and Prosthodontics, School of Dentistry at Araraquara, Universidade Estadual Paulista (UNESP), Araraquara 14801-903, SP, Brazil; 2Department of Morphology, Pediatric Dentistry and Orthodontic, School of Dentistry, São Paulo State University (UNESP), Araraquara 14801-903, SP, Brazil

**Keywords:** antimicrobial photodynamic therapy, inorganic nanoparticles, drug-delivery system, *Staphylococcus aureus*, *Escherichia coli*, systematic review, meta-analysis

## Abstract

Antimicrobial photodynamic therapy (aPDT) is a method that does not seem to promote antimicrobial resistance. Photosensitizers (PS) conjugated with inorganic nanoparticles for the drug-delivery system have the purpose of enhancing the efficacy of aPDT. The present study was to perform a systematic review and meta-analysis of the efficacy of aPDT mediated by PS conjugated with inorganic nanoparticles. The PubMed, Scopus, Web of Science, Science Direct, Cochrane Library, SciELO, and Lilacs databases were searched. OHAT Rob toll was used to assess the risk of bias. A random effect model with an odds ratio (OR) and effect measure was used. Fourteen articles were able to be included in the present review. The most frequent microorganisms evaluated were *Staphylococcus aureus* and *Escherichia coli*, and metallic and silica nanoparticles were the most common drug-delivery systems associated with PS. Articles showed biases related to blinding. Significant results were found in aPDT mediated by PS conjugated with inorganic nanoparticles for overall reduction of microorganism cultured in suspension (OR = 0.19 [0.07; 0.67]/*p*-value = 0.0019), *E. coli* (OR = 0.08 [0.01; 0.52]/*p*-value = 0.0081), and for Gram-negative bacteria (OR = 0.12 [0.02; 0.56/*p*-value = 0.0071). This association approach significantly improved the efficacy in the reduction of microbial cells. However, additional blinding studies evaluating the efficacy of this therapy over microorganisms cultured in biofilm are required.

## 1. Introduction

Antimicrobial-resistant infection is responsible for 5 million people’s deaths annually [1]. However, considering that 2 billion people live in countries with unsatisfactory diagnostic capacity, this deaths rates must be greater [2]. In addition, by the year 2050, antimicrobial-resistant pathogens are projected to kill 10 million people each year [3]. The misuse and overuse of antimicrobial agents are the main reasons related to the emergence and spread of antimicrobial-resistant infection [4]. In bacteria, the reduced permeability of antibiotics, the increased efflux pumps, changes in antibiotic targets by mutation, the modification and protection of targets, and the inactivation of antibiotics by hydrolysis or by the transfer of a chemical group are the main molecular mechanisms associated with bacterial resistance [5]. In fungi, the molecular mechanisms associated with acquired resistance are increased efflux pump activity, EGR, ERG11, CYP51, FKS mutations, and decreased membrane ergosterol activity [6]. 

In addition, the organization of microorganisms in biofilm provides an increase in the tolerance of microorganisms against conventional antimicrobial agents [7]. Biofilms are one of the most widely distributed and successful ways of life, which are defined as aggregates of microorganisms in which cells are frequently embedded in a self-produced matrix of extracellular polymeric substances [7,8]. In this scenario, the search for new therapeutic strategies for the inactivation of pathogenic microorganisms is extremely needed. 

Antimicrobial photodynamic therapy (aPDT) is a non-invasive form of treatment based on a combination of a photosensitizer, light with appropriate wavelength, and oxygen dissolved in the application site [9]. As a result of this combination, a photophysical and photochemical mechanism leads to a biological response via the production of reactive oxygen species (ROS), causing the destruction of the target [10]. In addition, the absence of development of microbial resistance to aPDT strengthens further studies for better optimization of this approach [11].

Despite significant results after aPDT application for the reduction of microbial load against many types of microorganisms [12,13,14,15,16,17], studies have been conducted to enhance the efficacy of aPDT [18]. In this context, a diverse type of nanoparticles has been used as a drug delivery system facilitating the photosensitizer uptake in microbial cells and improving bioavailability, solubility, permeability, and selectivity [19]. Inorganic nanoparticles are characterized by physical-chemical stability, good bioavailability, degradation resistance, low toxicity, and improved therapeutic action [20]. The inorganic nanoparticles are divided into metal oxides (iron oxide, zinc oxide, etc.) [21], semiconductors (graphene quantum dots), mesoporous silica, and metallic nanoparticles [22]. Thus, the present study aimed to perform a systematic review and meta-analysis of the effect of aPDT mediated by photosensitizers conjugated with inorganic nanoparticles to reduce microbial load. 

## 2. Materials and Methods

### 2.1. Protocol and Registration

The Preferred Reporting Items for Systematic Reviews (PRISMA) statement [23] was used to conduct the planning and running of the present systematic review and meta-analysis. In addition, the register was made in the Open Science Framework (OSF)—registration doi: 10.17605/OSF.IO/XNCGD.

### 2.2. Data Extraction and Research Question

The research question was based on the PICO strategy for systematic reviews, where P = microbial load evaluated in any type of study (e.g., in vitro, animal studies, clinical trials), I = aPDT associated with photosensitizers conjugated with inorganic nanoparticles, C = isolated therapy (aPDT) and O = reduction of microbial load. Secondary outcomes, for instance, changes in biofilm matrix, were also extracted. The present study aimed to answer the following focused question “Does the association between aPDT and inorganic nanoparticles increase the efficacy of the therapy for microbial control compared to aPDT only?” Further data on the name of the first author, the date of publication, study design, inorganic nanoparticle used, light dose, irradiation time (min), wavelength (nm), photosensitizer, pre-irradiation time, microorganism, culture type, sample size, and main outcomes were extracted from the articles included in this systematic review after the screening. 

### 2.3. Eligibility Criteria 

The inclusion criteria for this systematic review were the use of aPDT associated with inorganic nanoparticles for microbial control. There were no restrictions on the type of study design (e.g., the inclusion of in vitro and in vivo studies, observational human studies, and randomized clinical trials), language, and microorganisms. Studies that did not evaluate the isolated therapy (aPDT) as the control group, review articles, case reports, other modalities of treatment using inorganic nanoparticles, and aPDT combined with other modalities of treatment (e.g., photochemical therapy) were excluded. 

### 2.4. Search Strategy 

Firstly, two independent examiners were calibrated to perform all processes related to article selection. PubMed, Web of Science, Science Direct, Scopus, Cochrane Library, SciELO, and Lilacs databases were searched. For this end, we build the following search terms: (((antimicrobial photodynamic therapy) AND (Drug delivery system)) AND (Metallic nanoparticles)) AND (metal nanoparticles)((antimicrobial photodynamic therapy) AND (Drug delivery system)) AND (Metal oxides).((antimicrobial photodynamic therapy) AND (Drug delivery system)) AND (Carbon quantum dots)(((antimicrobial photodynamic therapy) AND (Drug delivery system)) AND (Mesoporous silica) AND (silica nanoparticles)

In addition, a manual search was also made, looking for articles to be included in the eligibility criteria proposed in the present systematic review and meta-analysis. Thus, relevant journals in the field of photodynamic therapy, nanoparticles, drug delivery systems, and ClinicalTrials.gov were searched. Based on the titles and abstracts, the two independent examiners selected and classified the articles as included or excluded from the review (Kappa score = 0.89). In this step, the Rayyan software was used for the selection process and for removing duplicate articles [24]. After the selection process was done, the data were extracted from the articles selected. Then, the articles were analyzed and discussed. Any disagreement during the process was solved by reaching a consensus before proceeding to the next steps. 

### 2.5. Qualitative Analysis 

In this step, OHAT Rob toll, adapted for in vitro studies [25,26], was used for the risk of bias assessment. In addition, for the question “Were there no other potential threats to internal validity?” this was considered as bias related to the sample size calculation, normality, and homoscedasticity evaluation and details about the inferential tests used in the statistical approaches [27]. 

### 2.6. Meta-Analysis and Quantitative Approaches

Meta-analysis and quantitative approaches were conducted by R software version 3.6.1 with the “META” package. A random effect model was used and the effect measure was the odds ratio (OR), configuring meta-analysis for binary outcomes. In the presence of sparse data, the “Peto” method was used. For n > 10, the publication bias was accessed by using a funnel plot, and for n < 10, a trim-and-fill analysis was conducted. The small-study effect and bias related to meta-analysis were accessed by the trim-and-fill method. High levels of heterogenicity were considered for I-squared > 50%. The presence of microbial cells after treatment was the outcome evaluated. The experimental group was formed by microbial cells that received aPDT with photosensitizer conjugated with inorganic nanoparticles and the control group was formed by microbial cells that received aPDT. When necessary, subgroup analysis was conducted. 

## 3. Results

### 3.1. Search Results

The article selection process is summarized in the flow diagram presented in Figure 1. The electronic search yielded 574 articles. Accordingly, 359 articles remained in the selection process. After title and abstract screening, 324 articles were excluded because they did not follow the eligibility criteria. A total of 35 articles were eligible for full-text evaluation. Subsequently, for full-text evaluation, 14 articles were included for qualitative analysis and 12 articles could be included in the meta-analysis.

### 3.2. Synthesis Results

The articles included in the present systematic review and meta-analysis ranged in publication data from 2015 to 2022 [28,29,30,31,32,33,34,35,36,37,38,39,40,41] (Table 1). Most of the studies have been performed in vitro studies. Only one study also performed an animal model assay [41]. The most frequently inorganic nanoparticle evaluated was the metal type, with the same proportion for gold [29,34,38] and for silver [31,36,39]. The second most common was mesoporous silica [28,30,31,32], followed by metallic oxides [37,40,41] and then carbon quantum dots [33,35]. In addition, the most frequently photosensitizer evaluated was the methylene blue [28,29,34,36,38], followed by curcumin [31,35,37], toluidine blue [30,33], phthalocyanines [39,40], rose Bengal [32], and chlorin e6 [41]. Except for curcumin, all photosensitizers evaluated have the same light absorption spectrum. The light dose and the pre-irradiation time showed great different values among the included articles. 

*Staphylococcus aureus* was the microorganism more frequently analyzed [29,30,33,36,37,38,39,40,41], followed by *Escherichia coli* [28,30,31,36], *Pseudomonas aeruginosa* [28,33], *Staphylococcus epidermis* [30], *Enterococcus faecalis* [34], *Aggregatibacter actinomycetemcomitans* [35], *Porphyromonas gingivalis* [35], *Prevotella intermedia* [35], and *Candida albicans* [32]. The microorganisms were mainly cultivated in suspension form [28,29,30,31,32,34,36,37,38,39,40,41]. Only five studies evaluated the microorganisms in biofilm [32,33,34,35,41]. Despite the different types of inorganic nanoparticles, significant results for microbial load reduction were reached for all studies included in the present systematic review, independently of the type of microorganism. Moreover, the photosensitizer conjugated with inorganic nanoparticles had significant results for inhibition of biofilm formation [33,34,35]. 

### 3.3. Risk of Bias Assessment 

For all articles included in the present systematic review, the main source of bias was related to blinding (Were research personnel blinded to the study group during the study? Can we be confident in the outcome assessment (including blinding of assessors?). Moreover, the lack of information about the conduction of the pilot study, sample size estimation, and statistical approaches (e.g., evaluation of outliers, verification of normal distribution, and homoscedasticity) were considered as other potential threats to internal validity. More details can be seen in Table 2. 

### 3.4. Meta-Analysis 

Meta-analyses were performed in 12 articles [29,30,31,32,33,34,35,36,37,38,40,41]. Only two studies could not be included in the quantitative approaches due to the lack of sample size reported. Firstly, we performed an overall meta-analysis, including the articles that evaluated microbial cells cultured in suspension (Figure 2). aPDT mediated by conjugated with inorganic nanoparticles has significant results comparing with aPDT for presence of microbial cells (OR = 0.19 [0.07; 0.67/*p*-value = 0.0019/I-squared = 0%). In addition, a prediction interval for the treatment effect in a future study from a random effects model was calculated (Figure 2A) [42], while a non-publication bias was detected (Figure 2B).

After, subgroup analysis was performed for *S. aureus* and *E. coli* (Figure 3). However, significant results for aPDT mediated by photosensitizers conjugated with inorganic nanoparticles was only reached out for *E. coli* (OR = 0.08 [0.01; 0.52]/*p*-value = 0.0081/I-squared = 0%) (Figure 3C). In addition, publication bias and consequently meta-analysis bias were detected by trim-and-fill analysis (OR = 0.06 [0.01; 0.30]/*p*-value = 0.0006/I-squared = 0%) (Figure 3D). 

Another subgroup analysis was performed using studies that evaluated methylene blue (MB), but no significant results were found—*p*-value = 0.1806 (Figure 4A). Despite the publication bias being detected and the meta-analysis bias being corrected, no significant result was found—*p*-value = 0.0692 (Figure 4B). One more meta-analysis was conducted for the viability of microbial cells cultured in biofilm; however, no significant result was found—*p*-value = 0.1270 (Figure 4A); non-publication and meta-analysis biases were also found (*p*-value = 0.1270) (Figure 4B). 

Lastly, subgroups meta-analyses were performed for Gram-positive (Figure 5A,B) and Gram-negative bacteria (Figure 5C,D) cultured in suspension and treated based on aPDT with photosensitizers conjugated with inorganic nanoparticles and aPDT only. Non-significant results were found for Gram-positive bacteria (Figure 5A) and no publication bias was detected (Figure 5). Nonetheless, significant results were found for Gram-negative bacteria (OR = 0.12 [0.02; 0.56]/*p*-value = 0.0071/I-squared = 0%) (Figure 5C) and after detection of publication and meta-analysis biases, the significant result remained (OR = 0.07 [0.02; 0.27]/*p*-value = 0.0001/I-squared = 0%) (Figure 5D). 

## 4. Discussion

In front of traditional antimicrobials agents, pathogenic microorganisms have developed different pathways in the field of persistence, tolerance, and resistance [5,6,43,44], leading to the emergence of multiple drug-resistant microorganisms and grating one of the major challenges in the medical field in the current society [45]. In addition, biofilm formation is one of the main reasons related to microbial resistance to drugs as bacteria as much as fungi have an innate tendency to cling onto biotic and abiotic surfaces and ensconce themselves in a self-produced matrix mainly containing extracellular polymeric substances (EPS), which are comprised of polysaccharides, extracellular DNAs, lipids, and proteins [7]. In this context, several approaches to combat biofilm have been developed, such as antimicrobial coating modification, antimicrobial peptides, quorum sensing inhibition, bioelectric and bioacoustics effects, and aPDT [43]. Multiple targets are involved and damaged in oxidative stress resulting from aPDT; microbial resistance is unlikely to occur and until now, it was not reported [11].

In this present systematic review and meta-analysis, we evaluated the use of photosensitizers conjugated with inorganic nanoparticles to enhance the aPDT efficacy. The scientific literature reported a wide type of natural and synthetic photosensitizers (PS) used as antimicrobial agents, such as chlorophyll, rose Bengal, crystal violet, methylene blue (MB), Toluidine blue, and psolaren [46]. In the present systematic review, we did not establish any restriction for the type of PS used by the included articles. 

The results observed in the present study are promising and can cooperate to better evaluation of this approach against pathogenic microorganisms. Most of the articles included in the systematic review highlighted significant improvements in the aPDT mediated by inorganic nanoparticles compared to aPDT only. These results comply with the overall results of the meta-analysis for microorganisms cultured in suspension and for *E. coli* also cultured in suspension. Although chemical and biochemical procedures were not evaluated in the present study, special attention should be given to this issue as the preparation of photomaterials with inorganic nanoparticles is complex and may modulate the results when applied in microbiology [46]. 

*E. coli* is a commensal bacterium of the gastrointestinal tract of humans as well as other mammals and birds. There are several pathogenic strains of *E. coli* that can cause diarrhea, enteric infections, urinary tract infections, meningitis, and septicemia [47,48]. In addition, the virulence mechanism related to this gram-negative bacterium involves adhesin expression, toxin secretion, iron acquisition factors, lipopolysaccharide structure, the presence of polysaccharide capsules, and invasion factors [47]. The biofilm formation of *E. coli* occurs in followed phases; reversible attachment, irreversible attachment, maturation, and dispersion [48]. In particular, the maturation phase is characterized by the final architecture and arrangement of the biofilm conferred by the matrix formation [49]. The matrix provides biofilm stability, promotes intercellular interaction, and enables the transport of nutrients and waste through the biofilms. Lastly, the matrix serves as a protective barrier against antimicrobial agents, antibodies, and host immune response [50,51]. The results of the present systematic review and meta-analysis highlighted that both approaches, aPDT with photosensitizers linked to inorganic nanoparticles and aPDT only, conferred significant results against *E. coli* viability cultured in a suspension model; however, the associated approach was significantly more efficient. Further studies should be designed to evaluate the efficacy of the therapy in *E. coli* cultivated in biofilm by the two options of treatment.

As opposed to *E. coli*, *S. aureus* is a Gram-positive opportunistic bacterium, catalase-positive, being considered a principal cause of nosocomial infections [52]. *S. aureus* is frequently found on mucosal surfaces (e.g., the nares, the throat, and the rectum) and in most regions of skin (e.g., axilla, groin, and perineum) [52]. These bacteria can be classified as Methicillin-sensitive *S. aureus* and Methicillin-resistant *S. aureus*, and both are relevant and associated with nosocomial infections [53]. Beyond skin and mucosal infections, *S. aureus* can be responsible for osteoarticular infections, medical devices-related infections, pneumonia, infective endocarditis, and bacteremia. Due to these broader aspects, *S. aureus* is associated with considerable morbidity, mortality, and economic costs for healthcare institutions [54]. The *S. aureus* biofilm development can be divided chronologically into four steps; attachment, multiplication, maturation, and detachment [55]. Similarly, as *E. coli* is in the maturation process of *S. aureus* biofilm, EPS is produced by covering the multicellular aggregation and granting survival aspects for the microorganism by hindering the function of the host immune system and acting as a multifunctional barrier against antimicrobial agents [55,56]. 

In the meta-analysis, no significant results were found for the viability of *S. aureus* in suspension when microorganisms were treated with aPDT mediated by photosensitizers linked to inorganic nanoparticles or aPDT only. The large standard deviation, as well as the small sample size leading to imprecision effects [57], may be the main reason for it. Therefore, these results should be interpreted carefully, as it is expected small sample sizes for in vitro studies compared to other types of studies (e.g., epidemiological studies and clinical trials), and consequently, in further meta-analysis studies involving a larger number of studies, significant results might be reached out. This explanation can also be stipulated for the other non-significant results found in the present meta-analysis, such as viability microorganisms cultured in biofilm and MB. 

Correlating the results found for *E. coli* and *S. aureus* with the results from Gram-positive and Gram-negative bacteria, we might hypothesize that photosensitizers conjugated with inorganic nanoparticles have an affinity with Gram-negative cell walls. In particular, the cell wall of this bacteria is mainly composed of lipopolysaccharide, while N-acetyl-muramic acid is the most common component of the Gram-positive cell wall [58]. 

MB can be used as a photosensitizer agent and is classified as a phenothiazine type with strong absorption between 630–680 nm [59]. Even though MB can be widely used as a photosensitizer for aPDT and there are no potential lethal adverse effects related to systemic administration, there are a few studies designed as clinical trials [59,60]. In addition, to enhance the efficacy of aPDT, nanoparticles can be conjugated with photosensitizers. Once nanoparticles can act as a drug-delivery system and consequently facilitate the internalization of photosensitizers, this approach confers to aPDT the lowest concentration of photosensitizers and the shortest light exposure time [61]. 

Nanoparticles based on metal and silica offer advantages over organic nanoparticles, such as having easy-to-control particle sizes, shape, porosity, and monodispersibility; however, these inorganic nanoparticles do not readily degrade in the biological system [62]. Among all metallic nanoparticles used in photodynamic therapy, gold nanoparticles are the most studied [63,64]. Gold and silver nanoparticles show a special optical phenomenon named localized surface plasmon resonance (LSPR) [63]. LSPR occurs when light interacts with conductive metallic nanoparticles that are smaller than the incident wavelength. Consequently, a rapid energy transfer from the metal surface to molecular O_2_ (oxygen molecule) with high efficiency occurs and forms ^1^O_2_ (singlet molecular oxygen), inducing aPDT even without the involvement of PS [63,64]. Silica nanoparticles are biocompatible, easy to produce, and show high photosensitizer loading capacities [65], and when loaded with antimicrobial compounds, silica nanoparticles have been shown to be capable of reversing antibiotic resistance [66]. In the present systematic review and meta-analysis, the gold and silver nanoparticles, as well as silica nanoparticles, were the inorganic materials most evaluated in conjugation with photosensitizers.

In all articles included in the systematic review, bias related to blinding was detected. Although blinding approaches are fundamental for clinical trials, in in vitro studies, their use is not often. However, the blinding application can eliminate and/or reduce biases related to effect sizes, turning the results more reliable [67]. Moreover, the absence of pilot studies and details about statistical approaches were considered potential threats for internal validity. The conduction of pilot studies can clearly identify a wide source of problems that directly affects the performance of the study and consequently affects the results found. In addition, the application of correct statistical tests, mainly based on the confirmation of normal distribution and homoscedasticity, are highly crucial for understanding the data [68]. 

Essentially, photosensitizers conjugated with inorganic nanoparticles enhance the effectivity of aPDT in the reduction of microbial load compared to aPDT mediated by classical photosensitizers. Thus, more studies should be planned to evaluate these approaches in other species of bacteria and fungi. Additionally, the evaluation of EPS after this therapeutic approach [69] will provide a wide window of understanding of the role of aPDT mediated by photosensitizers conjugated with inorganic nanoparticles in biofilm formation.

## 5. Conclusions

Photosensitizers conjugated with inorganic nanoparticles are an effective approach for the reduction of microbial load, especially for *E. coli* Gram-negative bacteria. Hence, this combined treatment has shown better results than compared to aPDT application only. However, additional blind studies are required to precisely evaluate the efficiency of the photosensitizers conjugated with inorganic nanoparticles as well as, and more studies are required to evaluate this approach in microorganisms cultured in biofilm.

## Figures and Tables

**Figure 1 pharmaceutics-14-02050-f001:**
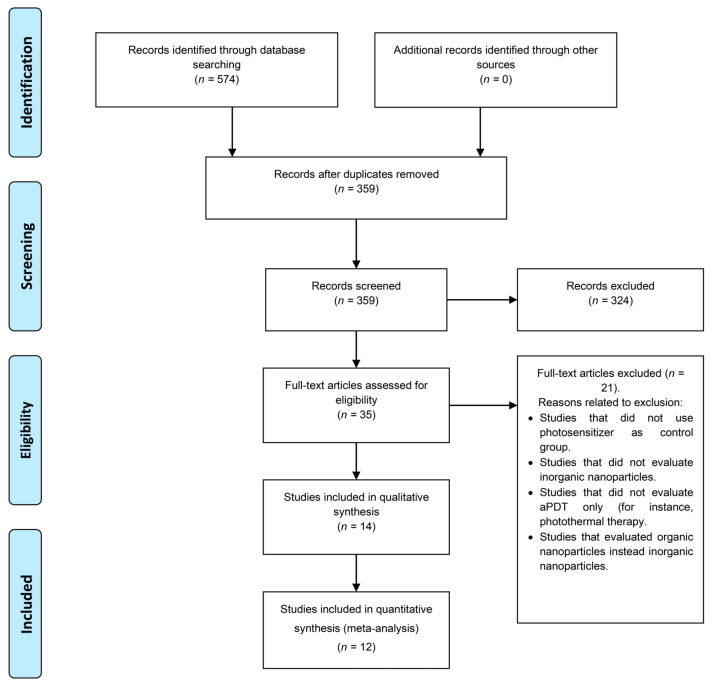
Flowchart based on the PRISMA statement.

**Figure 2 pharmaceutics-14-02050-f002:**
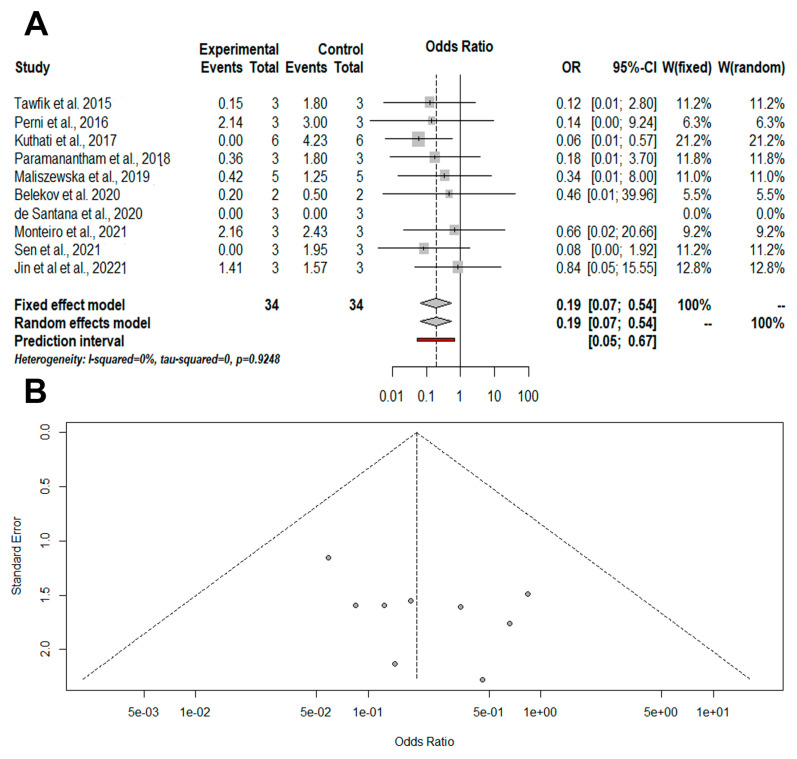
Illustration of meta-analysis and quantitative approaches. The experimental group was formed by aPDT mediated by photosensitizers conjugated inorganic nanoparticles and the control group was formed by aPDT only. (**A**) The results of the meta-analysis are illustrated in a forest plot for the overall viability of microbial cells. (**B**) Funnel plot analysis shows the absence of publication bias. OR = odds ratio; CI = confidence interval; W: weight [29,30,31,32,34,36,37,38,40,41].

**Figure 3 pharmaceutics-14-02050-f003:**
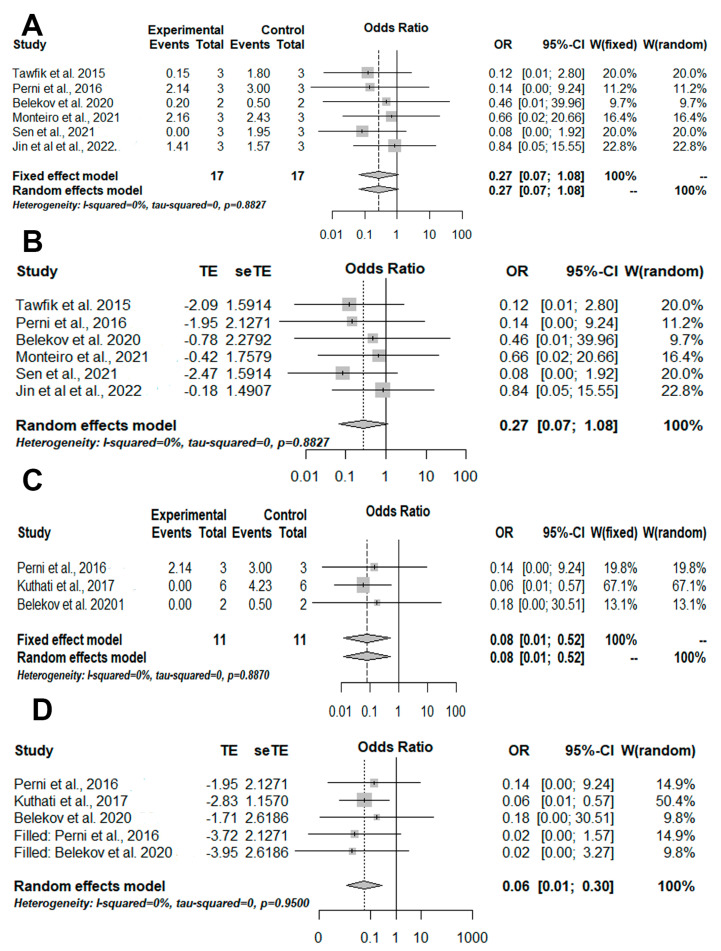
Illustration of meta-analysis and quantitative approaches. The experimental group was formed by aPDT mediated by photosensitizer conjugated inorganic nanoparticles and the control group was formed by aPDT only. (**A**) The results of the meta-analysis are illustrated in a forest plot for the viability of *S. aureus*. (**B**) Trim-and-fill results for the viability of *S. aureus* show the presence of publication biases. (**C**). The results of the meta-analysis are illustrated in a forest plot for the viability of *E. coli*. (**D**) Trim-and-fill results for the viability of *E. coli* show the presence of publication and meta-analysis biases. OR = odds ratio; CI = confidence interval; W: weight. TE = estimated mean; seTE = estimated standard deviation [29,30,31,36,38,40,41].

**Figure 4 pharmaceutics-14-02050-f004:**
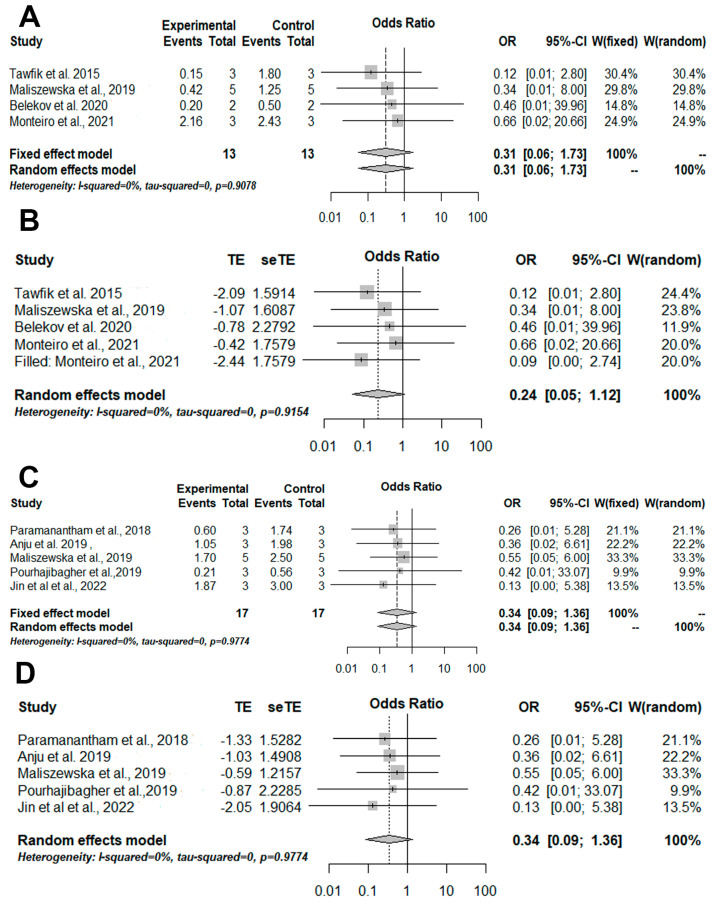
Illustration of meta-analysis and quantitative approaches. The experimental group was formed by aPDT mediated by photosensitizers conjugated inorganic nanoparticles and the control group was formed by aPDT only. (**A**) The results of meta-analysis illustrated in a forest plot for the viability of microbial cells cultured in suspension and treated with MB-mediated aPDT with or without photosensitizers conjugated with inorganic nanoparticles. (**B**) Trim-and-fill results for the viability of microbial cells cultured in suspension and treated with MB-mediated aPDT with or without photosensitizers conjugated with inorganic nanoparticles show the presence of publication and meta-analysis biases. (**C**) The results of meta-analysis illustrated in a forest plot for viability microbial cells cultured in biofilm. (**D**) Trim-and-fill results for viability microbial cells cultured in biofilm show the absence of publication bias. OR = odds ratio; CI = confidence interval; W: weight. TE = estimated mean; seTE = estimated standard deviation [29,32,33,34,35,36,38,41].

**Figure 5 pharmaceutics-14-02050-f005:**
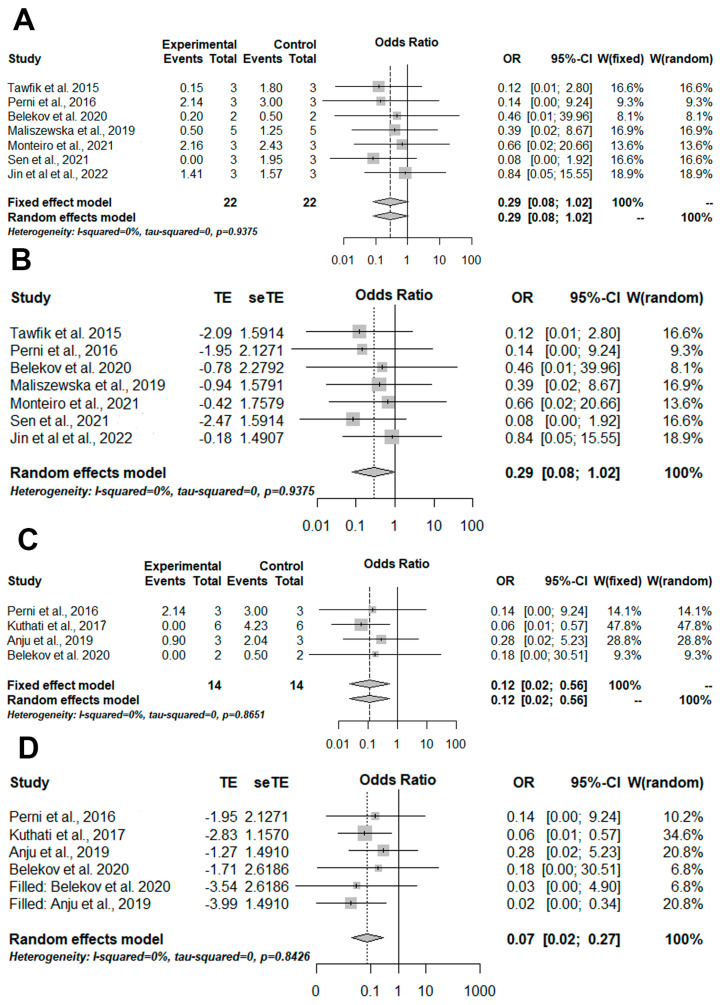
Illustration of meta-analysis and quantitative approaches. The experimental group was formed by aPDT mediated by photosensitizers conjugated inorganic nanoparticles and the control group was formed by aPDT only. (**A**) The results of the meta-analysis illustrated in a forest plot for the viability of Gram-positive bacteria cultivated in suspension. (**B**) Trim-and-fill results for the viability of Gram-positive bacteria cultured in suspension show the absence of publication and meta-analysis biases. (**C**) The results of the meta-analysis illustrated in a forest plot for viability Gram-negative bacteria cultivated in suspension. (**D**) Trim-and-fill results for Gram-negative bacteria cultivated in suspension presence of publication and meta-analysis biases. OR = odds ratio; CI = confidence interval; W: weight. TE = estimated mean; seTE = estimated standard deviation [29,30,31,33,34,36,38,40,41].

**Table 1 pharmaceutics-14-02050-t001:** Summary finds of data extraction from included articles.

Study (year)	Study Design	Inorganic Nanoparticle (np)	Light Dose	Irradiation Time	WaveLength	PhotoSensitizer	Pre-Irradiation Time	Microorganism	Culture Type	Sample Size	Outcomes
Planas et al., 2015 [28]	In vitro	Mesoporous Silica Nanoparticle (MSNP) modified with mannose sugars or amino groups	16 J/cm^2^	ND	652 nm	Methylene Blue (MB)	30 min	*Escherichia coli* *Pseudomonas* *aeruginosa*	Suspension	ND	*Colony forming units (CFU)* *E. coli* = reduction of 7 log_10_ using MB (10 µM) alone or associated with MSNP. *P. aeruginosa = reduction of 8 log_10_ using MB (10* *µM) alone or associated with MSNP targeting motifs with mannose sugars. Reduction of 5 log_10_ was observed using MB associated with MSNP targeting with amino groups.*
Tawfik et al., 2015 [29]	In vitro	Gold nanoparticles (AuNPs)	24 J/cm²	2 min	660 nm	Methylene blue (MB)	ND	*Staphylococcus aureus*	Suspension	3	*Cell viability after treatment* *S. aureus* Inhibition of 95% for MB+np and 40% after MB.
Perni et al., 2016 [30]	In vitro	Silica nanoparticle	ND	0.5 min 1 min 2 min 3 min	630 nm	Toluidine blue (TB)	ND	*Staphylococcus aureus* (MRSA), *Staphylococcus epidermidis,* and *Escherichia coli*	Suspension	3	*Colony forming units (CFU)* *E. coli* = Reduction of 2 log_10_ after 3 min of irradiation *S. epidermidis* = Reduction of 2 log_10_ after 2 min of irradiation, and after 3 min, the CFU fell below the limit detection. *S. aureus* = reduction of 2 log_10_ after 2 min of irradiation, and after 3 min, the CFU fell below the limit detection.
Kuthati et al., 2017 [31]	In vitro	Mesoporous silica (MSN) and silver nanoparticles (SNP)	72 J/cm^2^	300 seg	470 nm	Curcumin (Cur)	ND	*Escherichia coli*	Suspensions	6	*Cell viability (log CFU/mL)* -Cur: reduction of 6 log_10_ -Cur+np: total microbial reduction
Paramanantham et al., 2018 [32]	In vitro	Mesoporous silica (MSN)	50 mW	5 min	540 nm = Rose Bengal (RB) free 532 nm = MSN-RB	Rose Bengal (RB)	3 h	*Candida albicans*	Suspensions and biofilm		*Reductions in microbial suspension* -RB = 40.96 ± 2.71% -RB+np = 88.62 ± 3.4% *Reductions in microbial biofilm* -RB = 42.2 ± 2.6% -RB+np = 79.64 ± 3.05%
Anju et al., 2019 [33]	In vitro	Carbon nanotubes	58.49 J/cm²	3 min	630 nm	Toluidine blue (TB)	3 h	*Staphylococcus aureus* *Pseudomonas aeruginosa*	Biofilm	3	*Biofilm inhibition (crystal violet)* *S. aureus*: Reduction of 75% for TB+np and 47% after TB *P. aeroginosa:* Reduction of 70% for TB+np and 32% after TB *Cell viability inhibition (CFU/mL)* *S. aureus:* inhibition of 65% for TB+np and 34% after TB *P. aeroginosa:* inhibition of 58% for TB+np and 30% after TB *Inhibition of exopolysaccharide production * *S. aureus:* inhibition of 53% for TB+np and 30% after TB. *P. aeroginosa:* inhibition of 50% for TB+np and 27% after TB.
Maliszewska et al., 2019 [34]	In vitro	Gold nanoparticles (AuNPs)	55, 108, and 179 mW/cm^2^	5, 10, 15, 30, and 45 min	660 nm	MB	120 min	*Enterococcus* *faecalis*	Suspensions and biofilm	5	*Cell viability in suspension* *(log_10_CFU/mL)* -MB ~ 4.5 log_10_ of reduction -MB+np ~ 5.5 log_10_ of reduction *Cell viability in biofilm* *(log_10_CFU/mL)* -MB ~ 3 log_10_ of reduction -MB+np ~ 4 log_10_ of reduction
Pourhajibagher et al., 2019 [35]	In vitro	Graphene quantum dots (GQD)	60–80 J/cm²	1 min	435 ± 20 nm	Curcumin (CUR)	5 min	*Aggregatibacter actinomycetemcomitans*, *Porphyromonas gingivalis*, and *Prevotella* *intermedia*	Biofilm	3	*Cell viability inhibition* Reduction of 93% for GQD-CUR and 82% after CUR *Biofilm formation* Reduction of 76% for GQD-CUR and 61.3% after CUR
Belekov et al., 2020 [36]	In vitro	Silver nanoparticle	ND	5 min	660 nm	Methylene blue (MB)	ND	*Staphylococcus aureus* *Escherichia coli*	Suspension	2	*Colony forming units (CFU)* *S. aureus*: Reduction of 90 % for MB+np and 75% for MB *E. coli:* Reduction of 100 % for MB+np and 75% for after MB
de Santana et al., 2020 [37]	In vitro	Superparamagnetic iron oxide nanoparticles (SPIONPs)	3.12 J/cm²	29 seg	450 nm	Curcumin (CUR)	5 min	*Staphylococcus aureus*	Suspension	3	*Colony forming units (CFU)* SPIONPs + aPDT promoted the complete elimination of *S. aureus*. aPDT mediated by CUR promoted complete elimination using the same parameters.
Monteiro et al., 2021 [38]	In vitro	Gold nanoparticles (AuNPs)	125 mW; 12 J/cm^2^	192 seg	630 nm ± 20 nm	1,9-Dimethyl-Methylene Blue zinc chloride double salt (DMMB)	5min	*Staphylococcus aureus (MRSA)*	Suspensions	3	*Colony forming units*(log CFU/mL) -DMBMB: reduction of 9 log_10._ -DMMB-AuNPs: reduction of 8 log_10_
Sen et al., 2021 [39] (a)	In vitro	Silver nanoparticles	ND	80 min	680 nm	Phthalocyanines (complexes 2 and 3).	ND	*Staphylococcus aureus*	Suspension	ND	*Colony forming units (CFU)* 100% elimination of *S. aureus* employing light and the conjugate. 87.85% and 58.33% of reduction employing the Phthalocyanines complex numbers 2 and 3, respectively.
Sen et al., 2021 [40] (b)	In vitro	Nitrogen, sulfur co-doped GQDs (3@N,S-GQDs, 4@N,S-GQDs)	ND	80 min	687 and 685 nm	Phthalocyanines	ND	*Staphylococcus aureus*	Suspension	3	*Colony forming units (CFU)* ZnPC 3 + LED = 99.91% of reduction. ZnPC 4 + LED = 100% of reduction. Conjugated 3@N,S-GQDs + LED = 100% of reduction. Conjugated 4@N,S-GQDs + LED = 100% of reduction.
Jin et al., 2022 [41]	In vitro/ in vivo	Ce6@WCS-IONP	100 mW/cm^2^	15 min	660 nm	Chlorin e6	ND	*Staphylococcus aureus (MRSA)*	Suspension and Biofilm	3	*Colony forming units (log_10_ CFU/mL) (suspension)* Reduction of 4.25 log_10_ for Ce6@WCS-IONP and 3.8 log_10_ after Chlorin e6 *Cells in biofilm* Reduction of 37.5% for Ce6@WCS-IONP and no reductions after Chlorin e6 *Bacterial viability in an animal model* Reduction of 85% for Ce6@WCS-IONP and 50% after Chlorin e6

ND: not documented; min: minutes, seg: seconds; h: hour.

**Table 2 pharmaceutics-14-02050-t002:** Risk of bias assessment according to the OHAT criteria for in vitro studies.

	Question	Was Administered Dose or Exposure Level Adequately Randomized?	Was Allocation to Study Groups Adequately Concealed?	Were Experimental Conditions Identical across Study Groups?	Were Research Personnel Blinded to the Study Group during the Study?	Were Outcome Data Complete without Attrition or Exclusion from the Analysis?	Can We Be Confident in the Exposure Characterization?	Can We Be Confident in the Outcome Assessment (Including Blinding of Assessors?)	Were There No Other Potential Threats to Internal Validity?
Study									
Planas et al., 2015 [28]		**++**	**++**	**++**	**--**	**++**	**++**	**-**	**--**
Tawfik et al., 2015 [29]		**++**	**++**	**++**	**--**	**++**	**++**	**-**	**--**
Perni et al., 2016 [30]		**++**	**++**	**++**	**--**	**++**	**++**	**-**	**--**
Kuthati et al., 2017 [31]		**++**	**++**	**++**	**--**	**++**	**++**	**-**	**--**
Paramanantham et al., 2018 [32]		**++**	**++**	**++**	**--**	**++**	**++**	**-**	**--**
Anju et al., 2019 [33]		**++**	**++**	**++**	**--**	**++**	**++**	**-**	**--**
Maliszewska et al., 2019 [34]		**++**	**++**	**++**	**--**	**++**	**++**	**-**	**--**
Pourhajibagher et al., 2019 [35]		**++**	**++**	**++**	**--**	**++**	**++**	**-**	**--**
Belekov et al., 2020 [36]		**++**	**++**	**++**	**--**	**++**	**++**	**-**	**--**
de Santana et al., 2020 [37]		**++**	**++**	**++**	**--**	**++**	**++**	**-**	**--**
Monteiro et al., 2021 [38]		**++**	**++**	**++**	**--**	**++**	**++**	**-**	**--**
Sen et al., 2021 [39] (a)		**++**	**++**	**++**	**--**	**++**	**++**	**-**	**--**
Sen et al., 2021 [40] (b)		**++**	**++**	**++**	**--**	**++**	**++**	**-**	**--**
Jin et al., 2022 [41]		**++**	**++**	**++**	**--**	**++**	**++**	**-**	**--**

++: direct evidence of positive finding; -: indirect evidence of negative finding; --: direct evidence of negative finding.

## Data Availability

Not applicable.

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
