# Peer review of "Efficacy of Antimicrobial Photodynamic Therapy Mediated by Photosensitizers Conjugated with Inorganic Nanoparticles: Systematic Review and Meta-Analysis"

_pharmaceutics, 2022, doi:10.3390/pharmaceutics14102050_

Round 1

Reviewer 1 Report

In this work, Ana Claudia Pavarina et al. reported on the “Efficacy of antimicrobial photodynamic therapy mediate by 2 photosensitizers conjugated with inorganic nanoparticles: systematic review and meta-analysis.”

Their approach is innovative as such study is not usual for in vitro aPDT and thus interesting. However, the keywords used for publications selection are questionable. Indeed, the authors used “Metallic nanoparticles” when, very often, “metal nanoparticles” are used. The same kind of remark can be done for “mesoporous silica”. Indeed, just “silica nanoparticles or silica support” are usually used. As an example, we can cite the publication of Agnihotri, S. et al. “Synthesis and antimicrobial activity of aminoglycoside-conjugated silica nanoparticles against clinical and resistant bacteria”. (New J. Chem. 2015, 39, 6746–6755). I have not checked the eligibility of this paper according to Pavarina’s eligibility criteria and maybe is not, but I am pretty sure that we can find other paper. Can the authors check for other papers or explain their restrictive choice for keywords?

Also, may I suggest the authors add the Almeida review “Revisiting Current Photoactive Materials for Antimicrobial Photodynamic Therapy” (Molecules, 2018, 23(10):2424. doi: 10.3390/molecules23102424) in their introduction? Indeed, this quite recent review is totally in line with their subject.

Author Response

Reviewer 1

In this work, Ana Claudia Pavarina et al. reported on the “Efficacy of antimicrobial photodynamic therapy mediate by 2 photosensitizers conjugated with inorganic nanoparticles: systematic review and meta-analysis.”

Their approach is innovative as such study is not usual for in vitro aPDT and thus interesting. However, the keywords used for publications selection are questionable. Indeed, the authors used “Metallic nanoparticles” when, very often, “metal nanoparticles” are used. The same kind of remark can be done for “mesoporous silica”. Indeed, just “silica nanoparticles or silica support” are usually used. As an example, we can cite the publication of Agnihotri, S. et al. “Synthesis and antimicrobial activity of aminoglycoside-conjugated silica nanoparticles against clinical and resistant bacteria”. (New J. Chem. 2015, 39, 6746–6755). I have not checked the eligibility of this paper according to Pavarina’s eligibility criteria and maybe is not, but I am pretty sure that we can find other paper. Can the authors check for other papers or explain their restrictive choice for keywords?

Thank for your note. The keywords used were based on a recent article published in a prestigious journal (Silvestre ALP, Di Filippo LD, Besegato JF, de Annunzio SR, Almeida Furquim de Camargo B, de Melo PBG, Rastelli ANS, Fontana CR, Chorilli M. Current applications of drug delivery nanosystems associated with antimicrobial photodynamic therapy for oral infections. Int J Pharm. 2021 Jan 5; 592:120078. doi: 10.1016/j.ijpharm.2020.120078).  However, we performed another search in all databases evaluated using the keywords suggested “metal nanoparticles” and “silica nanoparticles” and no significant modifications for inclusion of new articles was reach out. In the present systematic review, we established a strict inclusion criterion, this is the reason of no others articles were able be included with the new keywords used.

In addition, the publication of Agnihotri, S. et al. “Synthesis and antimicrobial activity of aminoglycoside-conjugated silica nanoparticles against clinical and resistant bacteria”. (New J. Chem. 2015, 39, 6746–6755) did not evaluated the silica nanoparticles for aPDT application, so this publication was not included.

Also, may I suggest the authors add the Almeida review “Revisiting Current Photoactive Materials for Antimicrobial Photodynamic Therapy” (Molecules, 2018, 23(10):2424. doi: 10.3390/molecules23102424) in their introduction? Indeed, this quite recent review is totally in line with their subject.

Thank you for your note. We added the Almeida review (Revisiting Current Photoactive Materials for Antimicrobial Photodynamic Therapy” (Molecules, 2018, 23(10):2424. doi: 10.3390/molecules23102424) in the discussion of the present systematic review.

Reviewer 2 Report

The review is very interesting and presents the issue from original point of view.

Author Response

Reviewer 2.

The review is very interesting and presents the issue from original point of view.

Thank you for your note. We hope that the article could be published and help further articles in their development.
